# Keystroke Biometrics as a Tool for the Early Diagnosis and Clinical Assessment of Parkinson’s Disease

**DOI:** 10.3390/diagnostics13193061

**Published:** 2023-09-26

**Authors:** Wei-Min Liu, Che-Lun Yeh, Po-Wei Chen, Che-Wei Lin, An-Bang Liu

**Affiliations:** 1Department of Computer Science and Information Engineering, Advanced Institute of Manufacturing with High-Tech Innovations, National Chung Cheng University, Chiayi 621301, Taiwan; wmliu@cs.ccu.edu.tw (W.-M.L.); allen200168@gmail.com (C.-L.Y.); 2Department of Physical Medicine and Rehabilitation, Hualien Tzu Chi Hospital, Buddhist Tzu Chi Medical Foundation, Hualien 970473, Taiwan; drpwchen@gmail.com; 3Department of Biomedical Engineering, College of Engineering, National Cheng Kung University, Tainan 701401, Taiwan; lincw@mail.ncku.edu.tw; 4Department of Medicine, School of Medicine, Tzu Chi University, Hualien 970374, Taiwan; 5Department of Neurology, Hualien Tzu Chi Hospital, Buddhist Tzu Chi Medical Foundation, Tzu Chi University, Hualien 970473, Taiwan

**Keywords:** Parkinson’s disease, keystroke biometrics, early diagnosis, clinical assessments

## Abstract

(1) Background: Parkinson’s disease (PD) is the second most common neurodegenerative disease. Early diagnosis and reliable clinical assessments are essential for appropriate therapy and improving patients’ quality of life. Keystroke biometrics, which capture unique typing behavior, have shown potential for early PD diagnosis. This study aimed to evaluate keystroke biometric parameters from two datasets to identify indicators that can effectively distinguish de novo PD patients from healthy controls. (2) Methods: Data from natural typing tasks in Physionet were analyzed to estimate keystroke biometric parameters. The parameters investigated included alternating-finger tapping (afTap) and standard deviations of interkey latencies (IL_SD_) and release latencies (RL_SD_). Sensitivity rates were calculated to assess the discriminatory ability of these parameters. (3) Results: Significant differences were observed in three parameters, namely afTap, IL_SD_, and RL_SD_, between de novo PD patients and healthy controls. The sensitivity rates were high, with values of 83%, 88%, and 96% for afTap, IL_SD_, and RL_SD_, respectively. Correlation analysis revealed a significantly negative correlation between typing speed and number of words typed with the standard motor assessment for PD, UPDRS-III, in patients with early PD. (4) Conclusions: Simple algorithms utilizing keystroke biometric parameters can serve as effective screening tests in distinguishing de novo PD patients from healthy controls. Moreover, typing speed and number of words typed were identified as reliable tools for assessing clinical statuses in PD patients. These findings underscore the potential of keystroke biometrics for early PD diagnosis and clinical severity assessment.

## 1. Introduction

Parkinson’s disease (PD) is the second most common neurodegenerative disease in developed countries [1]. The disease is characterized by a range of manifestations, including bradykinesia, tremor, rigidity, and loss of postural balance, typically resulting from the progressive degeneration of dopaminergic neurons in the basal ganglia [2]. Current pharmacological therapies mainly aim to restore dopaminergic imbalance by using dopamine precursors, monoamine oxidase inhibitors, dopaminergic agonists, anticholinergic agents, and other drugs [3]. Levodopa effectively improves motor problems in patients with PD. However, the long-term administration of levodopa can lead to multiple unpredictable problems including dyskinesia, fluctuations in motor responses to levodopa for example, on–off or freezing, and impaired cognition. Furthermore, the use of current antiparkinsonian agents may aggravate the progression of PD [4,5]. According to these clinical observations, various preclinical and clinical trials are underway for the development of neuroprotective therapy, cell therapy, and anti-inflammatory treatments [6].

Studies have revealed that non-specific non-motor symptoms, such as constipation, sleep disorders, and depression, may manifest several years or decades before the diagnosis of PD [7]. Early and adequate therapeutic intervention can improve patients’ symptoms and their quality of life [2,8]. Additionally, suboptimal medication adherence, including underuse, overuse, and irregular use, is another common issue in clinical practice for PD. Due to motor complications, the excessive use of levodopa is another prevalent problem among patients with PD. Some patients with PD tend to overuse levodopa as it provides relief from their motor disability [9] Therefore, an objective and quantitative motor assessment for these patients under dopamine replacement therapy is important [10]. 

Motor dysfunctions in patients with PD are diverse and can be categorized into two major subtypes: tremor-dominant and non-tremor dominant. Patients with tremor-dominant PD typically exhibit more apparent clinical features and tend to have better outcomes compared to those with the non-tremor subtype. The diagnosis of non-tremor dominant PD can be challenging as the symptoms may not be as obvious as those in the tremor-dominant subtype. Additionally, motor dysfunctions such as rigidity and bradykinesia may not become apparent until the disease has already progressed. These symptoms can also be similar to those seen in other neurological disorders, leading to potential misdiagnosis or delayed diagnosis and treatment. The misdiagnosis of early PD has been reported to range from 40% to 70% [11,12]. Early diagnosis and the implementation of appropriate therapeutic strategies can significantly improve patients’ quality of life [13]. Researchers are investigating various biomarkers of PD, such as brain imaging, cerebrospinal fluid analyses, mutant genes, and genetic polymorphisms, in order to develop tools for early diagnosis, to monitor disease progression, and to evaluate the effectiveness of treatments [14]. 

As PD presents with a range of clinical features, including both motor and non-motor symptoms, a multidimensional assessment tool, the Unified Parkinson’s Disease Rating Scale (UPDRS), has been widely used in clinical evaluations and research [15]. The third part of the UPDRS (UPDRS-III), designed to assess motor functions in PD patients, has been used to determine the severity of PD and track the progression of the disease over time [16,17]. Given that tremor and locomotor dysfunctions, such as bradykinesia, festinating gait, and trunk imbalance, are the predominant motor manifestations of PD, kinesiological biomarkers such as gait and postural analyses, as well as tremography, are more relevant in a clinical setting than biological markers [18]. With the advancement of wearable and detection devices, both motor and non-motor signs of PD can be accurately assessed using several available devices [19,20]. Nevertheless, several non-invasive techniques have been studied. However, weaknesses and drawbacks exist, challenging the future applications of these methods [21]. Keystroke biometrics refers to the distinctive typing pattern and rhythm of an individual’s typing behavior. This type of biometric analysis examines keystroke dynamics, such as the timing between keystrokes, the duration of each keystroke, and the pressure applied to each key, to create a unique typing profile. Keystroke biometrics has been utilized for authentication purposes [22,23]. While keystroke biometrics have primarily found application in the domains of security and user authentication, their utilization in clinical settings is an emerging and relatively novel area of research. As a result, only a limited number of clinical studies have explored this biometric modality for various applications [24]. Some researchers have employed neural networks and sophisticated methods to analyze keystroke biometrics for the early diagnosis of Parkinson’s disease [25,26,27,28]. In 2016, Giancardo and colleagues proposed that keystroke biometrics could provide an early diagnostic indicator for PD [29]. The raw data from their study, including on UPDRS-III scores, typing speeds, the numbers of typed words, and letters typed in a natural typing task are freely available from PhysioNet [30]. In this study, we extracted and assessed these keystroke biometric datasets with the aim of identifying easy-to-use and sensitive indicators for the early diagnosis of PD and relevant parameters for assessing the severity and progression of PD. Our study innovatively applies keystroke biometrics to address critical clinical questions related to the diagnosis, severity assessment, and disease progression tracking of PD. As a result, we aim to introduce a novel, non-invasive, and user-friendly tool for clinicians and researchers in the field of neurodegenerative diseases. Furthermore, following the protocol outlined in the original article, this method can also be effectively applied in telemedicine.

## 2. Materials and Methods

### 2.1. Data Source and Subjects

We retrieved two datasets on a natural typing task from PhysioNet [30]. The demographic data, clinical status, experimental protocols, and keystroke biometric parameters assessed, including the neuroQWERTY index (nQi), single-key tapping (sTap), and alternating-finger tapping (afTap), were described in the previous publication [29]. In the “single key tapping” assessment, participants were instructed to rapidly press a single button for a duration of 60 s, initially using their dominant hand and subsequently with their non-dominant hand. The resulting score was defined as the average number of button presses achieved across both hands. In the “alternating finger tapping” test, participants were asked to alternately press two buttons, positioned approximately 25 cm apart, using their index finger. This test was conducted for both hands, and the ultimate score was defined as the average number of button presses performed across both hands. Four controls and five patients with early PD could not complete the afTap test. In the natural typing task, the participants were asked to type a folk tale on a standard word processor. They were required to type as they usually do, and they were allowed to freely correct any mistake. Each subject received a randomly selected folk tale to prevent the repetition of content and mitigate potential learning effects.

Two datasets are available in PhysioNet. The early PD dataset included 18 patients with early PD (Hoehn–Yahr stages I and II) without motor fluctuations and 13 healthy controls. The second dataset recruited 24 patients with de novo PD (i.e., newly diagnosed patients who were not yet taking antiparkinsonian agents) and 30 healthy subjects. Each of the enrolled subjects reported using a laptop or desktop computer for at least 30 min every day prior to the test. The participants performed a typing task twice. To avoid issues related to imprinting and adaptation, we used the dataset from the first typing test for analysis. In the current study, we assessed a total of 43 healthy controls from the two datasets.

### 2.2. Data Processing

The keystroke data were recorded as the time points when the keys were pressed and released during a natural typing task. Therefore, four time intervals were identified: holding latency (HL), the interval between release and press time points of a key; interkey latency (IL), the interval between the release time point of a keystroke and press time point of the consecutive keystroke; press latency (PL), the interval between two consecutive press time points; and release latency (RL), the interval between two consecutive release time points (Figure 1) [31].

### 2.3. Management of Outliers

Several data points in the datasets had unusually large values that were likely the result of unknown errors. These outliers were removed from the analysis, as were any time points that resulted in negative values. We also utilized Boxplots to identify outliers, defining extreme outliers as those falling outside the range that lay between Q1 − 3 × IQR and Q3 + 3 × IQR, where IQR = Q3 − Q1 [32]. However, due to the unusual distribution of outliers in the datasets, following this outlier definition would have led to the exclusion of too many potentially meaningful samples. To strike a balance, we conducted a thorough visual examination of the raw data, subject-by-subject, and optimized the 12 × IQR threshold to maximize the retention of meaningful samples.

### 2.4. Assessments of Fluctuation of the Time Intervals

To estimate fluctuations in time intervals, we employed a modified version of Lan’s method [33]. We calculated the standard deviations (SDs) of the natural log of the quotient of two consecutive time intervals. Our approach differed from the published method, which only included alphanumeric, symbol, and space bar keys. We included time points for all keystrokes. Standard deviations of holding latency, interkey latency, press latency, and release latency were conducted and remarked as HL_SD_, IL_SD_, PL_SD_, and RL_SD_.

### 2.5. Statistical Analyses

Data are expressed as means ± standard deviation (SD). The significance of the differences in demographic data, UPDRS-III, and keystroke biometric parameters between the healthy controls, patients with de novo PD, and patients with early PD were examined using a nonparametric ANOVA with Tukey post hoc test. The correlations between UPDRS-III and the keystroke biometric parameters were evaluated using Spearman’s rank correlation coefficients. In receiver operating characteristic curve (ROC curve), the 95% confidence interval (CI) for the area under the curve is an exact binomial CI. Paired comparisons of ROC curves were performed by using DeLong’s test as described previously [34]. All statistical analyses were performed using STATA software (Version 16.0 for Windows; STATA Corp. LLC, College Station, TX, USA). Statistical significance was defined as *p* < 0.05 for all tests.

## 3. Results

### 3.1. Demographic Data of the Participants

The datasets from PhysioNet did not provide demographic raw data on participants. However, the original article reported no significant differences in age, sex, or years of education between healthy controls and patients with de novo and early PD [29]. We made modifications to the grouping method used in the original publication and recruited healthy controls from the two retrieved datasets as a single control group. As shown in Table 1, there were no significant differences in age and average years of education among the three groups: healthy controls (mean age: 60.10 ± 10.20 years, mean education: 15.30 ± 5.20 years), patients with de novo PD (mean age: 61.40 ± 10.50 years, mean education: 15.50 ± 3.80 years), and patients with early PD (mean age: 55.90 ± 8.00 years, mean education: 14.83 ± 4.60 years). We removed some outlier keystroke time points from the datasets of each participant, and the percentages of removed time points were 0.56% ± 0.64% in controls, 0.40% ± 0.53% in patients with de novo PD, and 0.36% ± 0.42% in patients with early PD. The average duration after diagnosis was 1.60 ± 1.22 years for patients with de novo PD and 3.89 ± 1.23 years for patients with early PD. There was only one statistical significance in ‘Average duration after diagnosis’ of PD between the de novo PD and early PD participants, found using a nonparametric ANOVA with a Tukey post hoc test.

### 3.2. Comparisons of Clinical Severity and Keystroke Parameters in Controls and Patients with De Novo and Early PD 

Table 2 shows the comparisons of the UPDRS-III, nQi, and keystroke biometric parameters among the three groups. The healthy control group had significantly lower UPDRS-III, nQi, afTap, SD, IL_SD_, and RL_SD_ values compared to both the patients with de novo PD and the patients with early PD. There were no significant differences in any of these parameters between the patients with PD and the patients with de novo PD.

### 3.3. The Value of Keystroke Biometric Parameters for Early Diagnosis of PD

We used Receiver Operating Characteristic (ROC) curves to assess the ability of these typestroke parameters to differentiate de novo PD participants from the healthy controls. Notably, we focused our analysis on the discrimination between patients with de novo PD and healthy controls rather than the discrimination between the healthy controls and patients with either de novo PD or early PD in previous publications [29,33]. Figure 2a shows that nQi, afTap, SD, RL_SD_, and IL_SD_ are significantly effective in the early detection of PD. The area under the curve (AUC) values of afTap, IL_SD_, and RL_SD_ (0.751, 0.761, and 0.814, respectively) are higher than the previously reported methods of nQi and SD (0.729 and 0.716, respectively). Conversely, Figure 2b demonstrates that the AUC values of RL_SD_ and IL_SD_ decreased (0.560 and 0.625, respectively) while the AUC values of nQi and SD increased to 0.892 and 0.842. The AUC value of afTap was 0.785. However, there is no statistically significant difference between these groups.

We determined the optimal cut-off point for each parameter by using the maximal Youden indices [35] and evaluated the sensitivity and specificity of each parameter in distinguishing patients with de novo PD from the healthy controls. Table 3 shows the sensitivity and specificity for nQi, SD, afTap, IL_SD_, and RL_SD_ to differentiate de novo PD and early PD patients from the healthy controls.

### 3.4. The Correlations between Clinical Severity and Keystroke Biometric Parameters in Patients with PD 

Due to the small sample size, we used Spearman’s rank correlation coefficient to assess the correlations between the UPDRS-III and keystroke parameters. As depicted in Table 4, we observed strong negative correlations between typing speed and number of words typed with UPDRS-III in patients with early PD, with correlation coefficients of −0.757 and −0.733, respectively. In patients with de novo PD, the number of words typed and afTP exhibited moderate negative correlations with UPDRS-III, with correlation coefficients of −0.452 and −0.484, respectively. However, none of the other keystroke biometric parameters analyzed in this study displayed a significant correlation with UPDRS-III, either in patients with de novo PD or in patients with early PD.

## 4. Discussion

In this study, we analyzed the keystroke biometric parameters retrieved from the raw data of two datasets on PhysioNet, comparing healthy controls with patients with de novo or early PD. To ensure the accuracy of our results, we checked the datasets for errors and found a small number of outliers. Outliers can occur in any dataset, and they can significantly impact data analyses [36]. These outliers were not mentioned in previous studies [29,33]. Despite the presence of outliers, we found no significant differences in their occurrence among the three groups. Based on these findings, we concluded that removing these outliers would not affect the results of our data analysis. Our results were comparable to nQi and SD, as reported in the literatures [29,33].

To make an easier and reasonable assessment, we adapted Lan’s method by incorporating all time points for the typed keys. Our findings in Table 2 indicate significant differences between patients with PD and controls in UPDRS-III, nQi, afTap, SD, IL_SD_, and RL_SD_. In this study, our objective was to develop an assisting diagnostic tool for the early detection of de novo PD, distinct from previous studies that encompassed patients with either de novo or early PD [29,33]. Our results, shown in Figure 2a, reveal that the nQi, afTap, SD, IL_SD_, and RL_SD_ parameters had similar AUC values in differentiating de novo PD patients from the healthy controls. We used maximal Youden’s indices to define cut-off points for each parameter [35] and compared their sensitivity and specificity. nQi and SD had lower sensitivity rates (58% and 42%, respectively) than previously reported, possibly due to our strict grouping method. However, the afTap, IL_SD_, and RL_SD_ parameters showed higher sensitivity rates (83%, 88%, and 96%, respectively), making them more suitable for screening tests (Table 3). These parameters had lower specificity, but high sensitivity is crucial for a screening test as it helps to identify individuals who may require further diagnostic procedures [37]. Conversely, the AUC values of RL_SD_ and IL_S_D decreased while the AUC values of nQi and afTap increased when differentiating early PD patients from the healthy controls (Figure 2b). Additionally, the sensitivity of nQi, afTap, and SD increased, but RL_SD_ and IL_SD_ decreased significantly when distinguishing early PD patients from the healthy controls. Conversely, the specificity of these two parameters increased. This change could be attributed to the fact that subjects in the early PD group had been undergoing treatment for several years. Based on these findings, we propose that nQi and afTap are valuable indicators for detecting diagnosed PD. The decreased sensitivity of RL_SD_ and IL_SD_ suggests that these two parameters could serve as reliable markers for assessing the efficacy of PD treatment while their high specificity indicates their usefulness in confirming a PD diagnosis. These results highlight the novelty of our grouping methods compared to previous publications [29,33].

Lan and Giancardo suggested that HL could be a viable indicator for the early detection of PD. However, our current study found that the analyzed IL_SD_ and RL_SD_ were more effective than HL in detecting early PD. As per the definitions of keystroke biometric parameters, IL is the time interval between the release of one key and the subsequent press of another key, while RL is the interval between the release time points of two consecutive keystrokes. HL, on the other hand, measures the muscle activity involved in contracting and relaxing during typing. According to Lan’ s speculation, the SD represents the fluctuation of the time intervals [33]. While rigidity and bradykinesia are the primary symptoms of PD, they result in consistently increased muscle tone, such as in cogwheel or lead pipe rigidity, which would not impact the variation within HL. However, HL_SD_ may increase in other neurological conditions that cause inconsistent muscle tone such as spasticity resulting from stroke, dystonia, and dyssynergia in cerebellar disorders [38]. In contrast, IL refers to the time duration between releasing one key and pressing the next key. This motion involves a more intricate motor process including initiation, programming, and executing the movement of the motor machinery [12]. Based on this hypothesis, IL can serve as a more accurate indicator for diagnosing PD compared to HL. Furthermore, in the current typing test, IL can also be influenced by a person’s reading ability and vision, suggesting that IL may be a more sensitive measure than HL in the early diagnosis of PD with cognitive impairment. Additionally, IL_SD_ could potentially serve as a screening tool for cognitive and visual disorders as deviations from normal IL_SD_ values could indicate underlying neurological impairment. The afTap parameter measures consecutive random typing and is simpler than the natural typing task. On the other hand, the afTap parameter still involves a complex motor control system but is less affected by comprehension and vision. Therefore, the afTap parameter may be more relevant for the early detection of PD than the natural typing task.

Several kinesiological parameters, including gait, tremor, voice, and typing behavior, have been utilized to evaluate the severity of PD [39,40,41]. From a clinical perspective, assessments of gait, posture, and keystroke biometric parameters are relevant for monitoring the clinical course of patients with PD. UPDRS-III is a tool used to assess clinical severity, particularly in the context of disease progression and therapeutic efficacy [42,43]. The other aim of this study was to identify reliable keystroke biometric parameters that correlate with UPDRS-III and can be used to monitor the progression of PD. Interestingly, as shown in Table 3, only typing speed and number of words typed exhibited strong negative correlations with UPDRS-III as per Spearman’s rank correlation coefficient analyses. However, there were no significant differences in these two parameters between healthy controls and patients with PD, and they had poor ability to differentiate between de novo patients with PD and healthy controls. These findings suggest that typing speed and number of words typed could serve as reliable indicators for monitoring disease progression but are not suitable as screening assessments for PD diagnosis. The keystroke parameters discovered in this study have the potential to significantly improve diagnostic accuracy for PD. Furthermore, they can be seamlessly integrated into telemedicine protocols as originally proposed in the datasets [29]. Additionally, keystroke biometrics can be readily incorporated into smartphones to enable remote and longitudinal PD assessment, similar to applications in patients with multiple sclerosis [44]. Furthermore, the possibility of merging patient databases in a cloud service opens up avenues for remote diagnosis, assessment, extensive data analysis, and drug development. However, it is crucial to underscore the importance of data privacy and security when implementing keystroke biometrics in clinical practice, especially within the realms of cloud services and telemedicine.

## 5. Study Limitations

Our study is constrained by the relatively small dataset obtained from PhysioNet, which limits the generalizability of our findings. Additionally, the dataset lacks information on the participants’ drug histories, which could have confounded the results of our keystroke biometric analyses [28]. Meanwhile some risk factors, such as family history, physical activity, occupation hazards, and comorbidity, were not available in the datasets either. Moreover, the dataset encompasses heterogeneity in disease stages and treatment strategies, further impacting the robustness of our analysis. Therefore, this study aimed to straightforwardly assess the applicability of keystroke biometrics for detecting and assessing PD, drawing comparisons with previous studies.

## 6. Conclusions

This study suggests that the keystroke biometric parameters afTap, IL_SD_, and RL_SD_ are easy to assess and represent reliable tools for diagnostic screening for PD. Moreover, the typing speed and number of words typed may be useful for monitoring the clinical severity and progression of PD. These simple algorithms could be easily embedded in a laptop, tablet, or smartphone to enable the continuous remote monitoring of the clinical statuses of patients with PD and may help clinicians improve the accuracy of early diagnosis and the monitoring of disease progression.

## Figures and Tables

**Figure 1 diagnostics-13-03061-f001:**
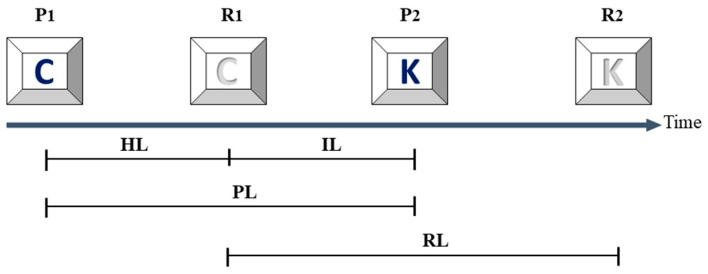
Definition of the typing intervals assessed in this study. The keystroke biometric parameters (dark: key pressed; grey: key released) assessed in the current study were based on the time points of press (P_1_, P_2_, P_3_, …P_n_) and release (R_1_, R_2_, R_3_, …R_n_) and defined as hold latency (HL), interkey latency (IL), press latency (PL), and release latency (RL) [31].

**Figure 2 diagnostics-13-03061-f002:**
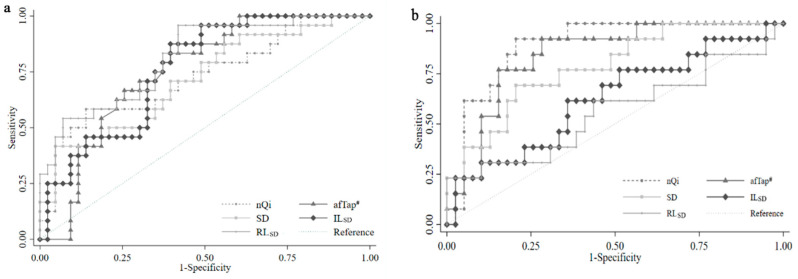
ROC curves showing the performance of nQi and keystroke biometric parameters to differentiate the de novo Parkinson’s disease patients from the healthy controls (**a**) and the early Parkinson’s disease patients from the healthy controls (**b**). nQi: neuroQWERTY index [29], SD: standard deviation of holding latency [33], RL_SD_: standard deviation of release latency, afTap: alternating-finger tapping, IL_SD_: standard deviation of interkey latencies. There were no statistically significant results as per DeLong’s test [34]. ^#^: Four healthy controls and five patients with early PD did not complete the afTap.

**Table 1 diagnostics-13-03061-t001:** Demographic data on the subjects assessed in this study.

Parameter	Healthy Controls (*n* = 43)	De Novo PD (*n* = 24)	Early PD (*n* = 18)
Age	60.10 ± 10.20	61.4 ± 10.50	55.90 ± 8.00
Sex (male)	17 (40%)	14 (58%)	10 (56%)
Average duration after diagnosis (years)	0	1.60 ± 1.22	3.89 ± 1.23 *
Average education (years)	15.30 ± 5.20	15.50 ± 3.80	14.83 ± 4.60
No. of outliers (%)	0.56 ± 0.64	0.40 ± 0.53	0.36 ± 0.42

Data are expressed as means ± SD. PD: Parkinson’s disease. No. of outliers: the percentage of outliers removed during data processing. * Significances of difference were determined as *p* < 0.05 by using nonparametric ANOVA with Tukey post hoc test between the groups.

**Table 2 diagnostics-13-03061-t002:** Clinical Severity and Keystroke Biometric Parameters.

Parameter	Healthy Controls (*n* = 43)	De Novo PD (*n* = 24)	Early PD (*n* = 18)
UPDRS-III (range)	1.92 ± 1.79 (0~6)	19.33 ± 6.70 * (7~36)	22.32 ± 8.69 ^†^ (11~40)
Typing speed (words/min)	112.34 ± 58.75	97.20 ± 42.53	98.86 ± 45.94
No. samples	1634.33 ± 793.04	1454.21 ± 497.72	1320.56 ± 581.98
nQi	0.06 ± 0.06	0.12 ± 0.10 *	0.14 ± 0.06 ^†^
sTap (msec)	170.85 ± 16.45	165.48 ± 24.24	159.42 ± 24.13
afTap ^#^ (msec)	128.99 ± 27.85	94.85 ± 23.54 *	96.33 ± 19.75 ^†^
SD	0.34 ± 0.08	0.41 ± 0.11 *	0.46 ± 0.14 ^†^
HL_SD_	0.50 ± 0.16	0.57 ± 0.15	0.53 ± 0.15
IL_SD_	1.15 ± 0.15	1.28 ± 0.11 *	1.27 ± 0.16 ^†^
PL_SD_	0.95 ± 0.13	1.01 ± 0.16	1.02 ± 0.22
RL_SD_	1.07 ± 0.12	1.23 ± 0.14 *	1.20 ± 0.19 ^†^

UPDRS-III: the third part of the Unified Parkinson’s Disease Rating Scale, no. samples: number of words typed, nQi: neuroQWERTY index [29], sTap: single-finger tapping, afTap: alternating-finger tapping, SD: standard deviation of holding latency [33], HL_SD_: standard deviation of holding latency based on our modified method, IL_SD_: standard deviation of interkey latency, PL_SD_: standard deviation of press latency, RL_SD_: standard deviation of release latency. * *p* < 0.05 in the comparison between the healthy controls and de novo PD participants; ^†^
*p* < 0.05 in the comparison between the healthy controls and early PD participants. Significances in difference were determined as *p* < 0.05 by nonparametric ANOVA with Tukey post hoc test. ^#^: Four healthy controls and five patients with early PD did not complete the afTap.

**Table 3 diagnostics-13-03061-t003:** Sensitivity and specificity of typestroke biometric parameters in de novo PD and early PD patients.

Parameter	De Novo PD		Early PD	
	Sensitivity	Specificity	Sensitivity	Specificity
nQi	58%	86%	94%	79%
afTap ^#^	83%	60%	92%	72%
SD	42%	95%	72%	77%
IL_SD_	88%	60%	44%	91%
RL_SD_	96%	58%	44%	100%

nQi: neuroQWERTY index [29], SD: standard deviation of holding latency [33], afTap: alternating-finger tapping, IL_SD_: standard deviation of interkey latency, SD: standard deviation of holding latency, RL_SD_: standard deviation of release latency. The cut-off points were defined by the maximal Youden indices [35]. ^#^: Four healthy controls did not complete the afTap.

**Table 4 diagnostics-13-03061-t004:** Correlation of Typestroke Parameters with UPDRS-III in Patients with Parkinson’s Disease.

Parameter	De Novo PD	Early PD
	(*n* = 24)	(*n* = 18)
Typing speed	−0.371	−0.757 ^†^
No. samples	−0.452 *	−0.733 ^†^
nQi	0.243	0.353
sTap	−0.112	0.077
afTap	−0.484 *	−0.095
SD	0.089	0.212
HL_SD_	−0.005	−0.019
IL_SD_	−0.09	−0.283
PL_SD_	0.274	0.347
RL_SD_	−0.085	−0.144

No. samples: number of words typed, nQi: neuroQWERTY index [29], sTap: single-finger tapping, afTap: alternating-finger tapping, SD: standard deviation of holding latency [33], HL_SD_: standard deviation of holding latency based on our modified method, IL_SD_: standard deviation of interkey latency, PL_SD_: standard deviation of press latency, RL_SD_: standard deviation of release latency. * *p* < 0.05 for the significant difference between the healthy controls and patients with de novo PD, ^†^
*p* < 0.05 for the significant difference between the healthy controls and patients with early PD. Significances of difference were determined as *p* < 0.05 by using nonparametric Spearman rank correlation.

## Data Availability

Data used in this investigation are available under the ODC Public Domain Dedication and from the PhysioNet repository at https://archive.physionet.org/physiobank/database/nqmitcsxpd/?C=S;O=A (accessed on 22 September 2023).

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
