# Peer review of "Keystroke Biometrics as a Tool for the Early Diagnosis and Clinical Assessment of Parkinson’s Disease"

_diagnostics, 2023, doi:10.3390/diagnostics13193061_

Round 1
Reviewer 1 Report
The article explores the use of keystroke biometrics as a simple tool for the early diagnosis of Parkinson's disease (PD). To this end, specific biometric parameters were estimated for two datasets collected in a previous experimental study involving early-stage PD subjects, de-novo diagnosed PD subjects, and healthy controls. The analysis covers specific parameters related to keystroke and keystroke release events, as well as typing speed and number of words typed. The results, in terms of ROC curves (specificity and sensitivity) and correlations with UPDRS-III scores, suggest the potential of these metrics in early diagnosis and monitoring of PD.
The methodology used to discriminate healthy controls from early PD is interesting because of its simple administration. However, the study has some weak and unclear points that need to be improved before publication.
The authors should better address the following points:
1) Introduction: The section mainly focuses on the characteristics of Parkinson's disease, while it does not mention studies related to PD assessment using other types of technologies, which, on the contrary, have been widely investigated in recent years for PD severity staging, early diagnosis, assessment, and monitoring. The authors should improve the introduction by including references in this regard.
2) Introduction: The section needs more references to other studies using the same approach (keystroke biometrics) as an assessment tool in clinical applications. The authors should introduce further studies in this regard.
3) Introduction: At the end of the section, the authors should better explain the objectives and innovation of their study compared to the state-of-the-art.
4) Line 97: For clarity, it would be better to briefly describe the parameters even though they were described in [18]. The parameter "sTap," investigated by the study, should also be described.
5) Line 99: More information about the "folk tale" should be provided, such as the number of words and characters. Was the "folk tale" the same for all participants in the two datasets?
6) Line 100-101: Is there information (per subject) on the number of corrections in the two databases? How might the estimated parameters be affected by the correction of mistakes?
7) Line 128: How was the "100-second" threshold defined? The authors should provide more information about the discharged outliers, specifying both the number of “good” samples and the number of outliers.
8) Line 130: Lan's method needs a reference.
9) Lines 132-133: The authors should better explain the differences of the proposed approach from [21] to clarify the sentence.
10) Lines 137-138: It needs to be clarified whether the statistical analysis of demographic data to estimate the significance of differences was performed or not. This sentence contrasts with lines 151-152, 160-162 and 172, where the authors state that statistical analysis of demographic data was not performed due to lack of raw data.
11) Line 140: The authors should justify the use of nonparametric tests.
12) Table 1: Does the percentage of outliers refer to both datasets? How many valid samples were available in the two datasets?
13) Line 172: If statistical analysis was not performed on different demographics (age, sex, duration, and education), it cannot be said that there are no statistical differences between the three groups (lines 155-159). However, the values in Table 1 suggest a statistical difference in some parameters of early PD, at least about age (compared with de-novo and healthy controls) and duration (compared with de-novo). The authors should provide more details on this point. Why was reference [25] cited?
14) Table 2: It is not common practice to examine healthy controls using UPDRS-III. Since UPDRS data are available, it would be interesting also to include the range (minimum and maximum scores) for the three groups.
15) Table 2: Curiously, the typing speed and number of samples parameters do not show a statistical difference. What are the p-values resulting from the statistical analysis?
16) Table 2: The units of measurement of the parameters should be specified.
17) Table 2: The use of the symbol "**" for the early PD column could be misleading to the reader. Commonly, the use of "**" for statistical analysis results refers to the p-value < 0.01, whereas, according to row 198, the p-value is < 0.05 (as for the De-novo PD column). It would be better to use "*" for the early PD column as well.
18) Lines 185-187: Parameter names should be the same throughout the document (SD should be subscript).
19) Lines 194-197: ROC analysis should also be provided for the early PD group for consistency with the correlation analysis in the next paragraph.
20) Line 207: The phrase "SD: standard deviation of retention latency" is repeated.
21) Line 209: The "SD" for the RLSD and ILSD parameters should be a subscript.
22) Line 212: A reference for Youden indices should be added.
23) Lines 215-216: It is unnecessary to repeat the specificity and sensitivity values when they are given in Table 3.
24) Table 3: In the caption, it would be better to point out that the table refers to the data for the de-novo PD group. Why not also include the same data for early PD (for consistency with the correlation analysis)?
25) Line 225: Authors should indicate the sample size used for the analysis (preferably in the Materials and Methods section).
26) Table 4: The Healthy Controls column could be unnecessary because assessing UPDRS in healthy controls is not common. In addition, typing speed and number of samples are correlated with UPDRS-III scores. In contrast, they are not statistically different in Table 2. The authors should discuss this conflicting aspect in more detail.
27) Table 4: See item 17.
28) Line 236: Why was [19] given instead of [21] as in Table 2? The same applies to Table 3. What is the correct reference for SD?
29) Lines 245-247: Since both de-novo and early PD are mentioned, all analyses should be performed, and results should be provided for both groups of Parkinsonian subjects.
30) Lines 255-256: Table 1 is mentioned, but it should be Table 2. In addition, stating a significant difference between healthy controls and PD patients for UPDRS-III is not relevant. It would be more interesting to show a statistical difference between de-novo PD and early PD regarding UPDRS scores.
31) Lines 257-259: This sentence seems incomplete.
32) Lines 304-306: This sentence seems at odds with lines [30-31] in the abstract. However, the results (statistical analysis versus correlations) for typing speed and number of samples also seem at odds.
33) Line 308: What do the authors mean by "heterogeneous clinical evaluation"? What do they mean by "well-controlled cohort"? These two sentences could invalidate the proposed method and the results presented if not adequately clarified.
34) References: The authors should include more recent references.
English is correct in form and punctuation. No major errors were found, only some typos need to be corrected.
Author Response
Reviewer 1:
Comments and Suggestions for Authors
The article explores the use of keystroke biometrics as a simple tool for the early diagnosis of Parkinson's disease (PD). To this end, specific biometric parameters were estimated for two datasets collected in a previous experimental study involving early- stage PD subjects, de-novo diagnosed PD subjects, and healthy controls. The analysis covers specific parameters related to keystroke and keystroke release events, as well as typing speed and number of words typed. The results, in terms of ROC curves (specificity and sensitivity) and correlations with UPDRS-III scores, suggest the potential of these metrics in early diagnosis and monitoring of PD.
The methodology used to discriminate healthy controls from early PD is interesting because of its simple administration. However, the study has some weak and unclear points that need to be improved before publication.
The authors should better address the following points:
We would like to express our gratitude for the reviewer's meticulous examination and insightful comments. Below, we provide our responses and the corresponding corrections.
1. Introduction: The section mainly focuses on the characteristics of Parkinson's disease, while it does not mention studies related to PD assessment using other types of technologies, which, on the contrary, have been widely investigated in recent years for PD severity staging, early diagnosis, assessment, and The authors should improve the introduction by including references in this regard.
Reply: Thank you for the kindly remind. We have added a comprehensive review article about the recent assessing techniques and devices for motor and non-motor signs of PD in the revised manuscript. Lines 81-84
2. Introduction: The section needs more references to other studies using the same approach (keystroke biometrics) as an assessment tool in clinical The authors should introduce further studies in this regard.
Reply: We appreciate the reviewer's feedback regarding the need for additional references to studies employing keystroke biometrics in clinical applications in the introduction section. While keystroke biometrics have primarily found application in the domains of security and user authentication, their utilization in clinical settings is an emerging and relatively novel area of research. As a result, only a limited number of clinical studies have explored this biometric modality for various applications. In response to this valuable comment, we have revised the introduction section of the manuscript to emphasize the novelty and evolving nature of keystroke parameters' clinical applications, particularly in the context of neurodegenerative diseases. Lines 88-94
3. Introduction: At the end of the section, the authors should better explain the objectives and innovation of their study compared to the state-of-the-art.
Reply: We appreciate the reviewer's feedback regarding the need for a clearer explanation of the objectives and innovation of our study compared to the current state-of-the-art. In our revised manuscript, we explicitly stated that our study aims to advance the field of early Parkinson's disease (PD) diagnosis by exploring the untapped potential of keystroke biometrics, a relatively novel approach in clinical applications. We have rewritten the manuscript. Lines 100-105
4. Line 97: For clarity, it would be better to briefly describe the parameters even though they were described in [18]. The parameter "sTap," investigated by the study, should also be described.
Reply: We added a brief description in the revised manuscript Lines 111-118
5. Line 99: More information about the "folk tale" should be provided, such as the number of words and characters. Was the "folk tale" the same for all participants in the two datasets?
Reply: Thank you for the critical comment. No, the folk tales were selected randomly. Therefor the number of words and characters were not the same. We added a paragraph in the revised manuscripts and describe the purpose of the original study design. Lines 122-124
6. Line 100-101: Is there information (per subject) on the number of corrections in the two databases? How might the estimated parameters be affected by the correction of mistakes?
Reply: Thank you for the valuable comment. The detail typing information, including corrections, can be found in the original datasets on the website: https://archive.physionet.org/physiobank/database/nqmitcsxpd/?C=S;O=A. In the current study, we treated typing and correcting as similar behaviors, comprising initiation, programming, and executing the movement of the motor machinery. Correction typing involves more time, including speech checking and recomposition before initiation, which may affect the final results. However, there are only a few corrections in the datasets. Moreover, numbers of correction may reflect the cognitive function of the subjects. Overall, the typing task would be a reliable assessment of PD beside motor assessment. The original article and the comparative article did not take into account the effects of correction in their research.
7. Line 128: How was the "100-second" threshold defined? The authors should provide more information about the discharged outliers, specifying both the number of “good” samples and the number of outliers.
Reply: We appreciate the valuable comment. In response, we've made specific improvements to address outlier management in our Initially, we set a threshold of 100 seconds to identify and remove extreme outliers based on empirical and visual inspection. In the revised manuscript, we utilized the Box plots in the following figures to identify outliers. In the original literature the extreme outliers are defined as those falling outside the range between Q1 - 3xIQR and Q3 + 3xIQR, where IQR = Q3 - Q1 [1]. From the figure we can see that most latency values falling between the lower quartile Q1 and the upper quartile Q3 are in the interval [0,1] second. However, due to the unusual distribution of outliers in the datasets (extreme values more than ±1000) seconds, following this outlier definition would have led to the exclusion of too many potentially meaningful samples. To strike a balance, we conducted a thorough visual examination of the raw data, subject-by-subject, and optimized the 12xIQR threshold to maximize the retention of meaningful samples. Consequently, the final data used for analysis are similar, and the results and statistical significance are identical to those obtained using the previous method. We have rewritten the method in the revised manuscript. Lines 149-155.

For example, here are the holding latencies for ID 60 (healthy control) with extreme outliers (A) and the refined data used for analysis after outlier removal (B).

8. Line 130: Lan's method needs a reference.
Reply: We have added the cited reference.
9. Lines 132-133: The authors should better explain the differences of the proposed approach from [21] to clarify the sentence.
Reply: We try to clary our method as the highlighted sentence To estimate fluctuations in time intervals, we employed a modified version of Lan's method [2]. We calculated the standard deviations (SD) of the natural log of the quotient of two consecutive time intervals. Our approach differed from the published method, which only included alphanumeric, symbol, and space bar keys. We included time points for all keystrokes. Standard deviations of holding latency, interkey latency, press latency and release latency were conducted and remarked as HLSD, ILSD, PLSD and RLSD. Lines 159-160
10. Lines 137-138: It needs to be clarified whether the statistical analysis of demographic data to estimate the significance of differences was performed or This sentence contrasts with lines 151-152, 160-162 and 172, where the authors state that statistical analysis of demographic data was not performed due to lack of raw data.
Reply: Thank you for the comments. We rewrote the manuscript and used one-way ANOVA with Tukey post-hoc test to examine the differences in parameters between the groups. All the demographic data were examined. Only the durations after the diagnosis of PD between the de-novo PD and early PD groups showed significant differences (Table 1).
11. Line 140: The authors should justify the use of nonparametric tests.
Reply: Thank you for pointing out the inappropriate statistical method. In deed the statistical analyses were conducted by nonparametric ANOVA with Tukey post- hoc test. They are corrected in the revised manuscript.
12. Table 1: Does the percentage of outliers refer to both datasets? How many valid samples were available in the two datasets?
Reply: Yes, the percentage of outliers was consistent across both datasets. After excluding the outliers, the valid samples ranged from 306 to 3652. While the number of typed words varied significantly among subjects, the numerical distribution of intervals within each group remained homogeneous. Following the outlier removal method mentioned in response to comment 7, the distribution of HL became more condensed. Specifically, the HLs were 110 ± 30 msec, 130 ± 40 msec, and 150 ± 40 msec in the healthy control, de-novo PD, and early PD groups, respectively. Additionally, Table 2 demonstrates the homogeneity of HLSD, ILSD, PLSD, and RLSD across the groups.
13. Line 172: If statistical analysis was not performed on different demographics (age, sex, duration, and education), it cannot be said that there are no statistical differences between the three groups (lines 155-159). However, the values in Table 1 suggest a statistical difference in some parameters of early PD, at least about age (compared with de-novo and healthy controls) and duration (compared with de- novo). The authors should provide more details on this point. Why was reference [25] cited?
Reply: Thank you for pointing out the inappropriate statistical method. In deed the statistical analyses were conducted by nonparametric ANOVA with Tukey post- hoc test. In Table 1, all the demographic parameters including age, sex, average duration of disease, and average education were analyzed. They are corrected in the revised manuscript. Table 1 and Lines 190-193
14. Table 2: It is not common practice to examine healthy controls using UPDRS-III. Since UPDRS data are available, it would be interesting also to include the range (minimum and maximum scores) for the three groups.
Reply: The UPDRS-III of the healthy control were retrieved from the previous publication [3]. We have added the ranges of UPDRS-III of each group in the revised Table 2.
15. Table 2: Curiously, the typing speed and number of samples parameters do not show a statistical difference. What are the p-values resulting from the statistical analysis?
Reply: Due to the heterogeneous distribution of the original data, no statistical significance was observed between healthy controls and patients with de-novo PD or early PD in both typing speed and the number of typed words. The P values were as follows: 0.273 between healthy controls and de-novo PD for typing speed, 0.389 between healthy controls and early PD for typing speed, 0.318 between healthy controls and de-novo PD for the number of typed words, and 0.135 between healthy controls and early PD for the number of typed words.
16. Table 2: The units of measurement of the parameters should be specified.
Reply: Thank you for the comment. We have added units of measurement of the parameters in the revised Table 2.
17. Table 2: The use of the symbol "**" for the early PD column could be misleading to the reader. Commonly, the use of "**" for statistical analysis results refers to the p-value < 01, whereas, according to row 198, the p-value is < 0.05 (as for the De-novo PD column). It would be better to use "*" for the early PD column as well.
Reply: Thank you for your correction. We use these two symbols to mark the comparison between healthy controls and patients with de-novo PD and healthy controls and patients with early PD respectively. In the revised version we use ¶ instead of ** to enhance clarity and consistency.
18. Lines 185-187: Parameter names should be the same throughout the document (SD should be subscript).
Reply: Thank you for the careful checkout. The parameters have been corrected.
19. Lines 194-197: ROC analysis should also be provided for the early PD group for consistency with the correlation analysis in the next paragraph.
Reply: We appreciate the reviewer’s valuable comment and have added an ROC analyses for the early PD group. There are some interesting differences compared to the ROC for the patients with de-novo PD. Therefore, we have also included more discussions on these new results in the revised manuscript. Figure 2, Table 3, Lines 218-228, 288-310
20. Line 207: The phrase "SD: standard deviation of retention latency" is repeated.
Reply: the repeat has been omitted in the revised manuscript.
21. Line 209: The "SD" for the RLSD and ILSD parameters should be a subscript.
Reply: the typos have been corrected.
22. Line 212: A reference for Youden indices should be added.
Reply: The reference has been added.
23. Lines 215-216: It is unnecessary to repeat the specificity and sensitivity values when they are given in Table 3.
Reply: This paragraph has been revised.
24. Table 3: In the caption, it would be better to point out that the table refers to the data for the de-novo PD group. Why not also include the same data for early PD (for consistency with the correlation analysis)?
Reply: Similar to the response to comment 19, we have added an ROC curve for patients with early PD. We have included the new results in our manuscript and revised Table 3 as per the reviewer’s respected suggestion.
25. Line 225: Authors should indicate the sample size used for the analysis (preferably in the Materials and Methods section).
Reply: Thank you for the comment. Yes, the number of subjects in each group has been specified in the Materials and Methods section.
26. Table 4: The Healthy Controls column could be unnecessary because assessing UPDRS in healthy controls is not common. In addition, typing speed and number of samples are correlated with UPDRS-III In contrast, they are not statistically different in Table 2. The authors should discuss this conflicting aspect in more details.
Reply: Thank you for the valuable suggestion. Table 4 has been revised. Because of heterogeneous distribution of typing speed and number of word typed, there is no statistical significance between the groups.
27. Table 4: See item 17.
Reply: Similar to the response to comment 17, we have revised Table 4.
28. Line 236: Why was [19] given instead of [21] as in Table 2? The same applies to Table What is the correct reference for SD?
Reply: We appreciate the reviewer’s careful checkup. We have corrected the typos in the revised manuscript.
29. Lines 245-247: Since both de-novo and early PD are mentioned, all analyses should be performed, and results should be provided for both groups of Parkinsonian subjects.
Reply: Thank you for the suggestion. We have revised the manuscript.
30. Lines 255-256: Table 1 is mentioned, but it should be Table In addition, stating a significant difference between healthy controls and PD patients for UPDRS-III is not relevant. It would be more interesting to show a statistical difference between de-novo PD and early PD regarding UPDRS scores.
Reply: Thank you for the suggestion. We have revised the manuscript.
31. Lines 257-259: This sentence seems incomplete.
Reply: We have omitted this sentence.
32. Lines 304-306: This sentence seems at odds with lines [30-31] in the abstract. However, the results (statistical analysis versus correlations) for typing speed and number of samples also seem at odds.
Reply: Regarding the concern raised in the comment, we would like to clarify that in the abstract (Lines 30-31), we stated that keystroke biometrics, afTap, ILSD, and RLSD are useful for screening tests of PD, as shown in Table 3. Additionally, we highlighted that typing speed and the number of words typed can serve as indicators for the severity of early PD patients, as demonstrated in Table 4. While it's true that, as continuous variables, typing speed and the number of samples did not show a statistically significant difference between the de-novo PD and early PD groups, they do exhibit a significant correlation with UPDRS-III when examined using the Spearman rank correlation coefficient test (Table 4). Therefore, there is no conflict between these two sets of results.
33. Line 308: What do the authors mean by "heterogeneous clinical evaluation"? What do they mean by "well-controlled cohort"? These two sentences could invalidate the proposed method and the results presented if not adequately clarified.
Reply: We appreciate the valuable comment. We have rewritten the “Study Limitations”. Lines 360-368.
34. References: The authors should include more recent references
Reply: We have added more recent references in the revised manuscript.
Comments on the Quality of English Language
English is correct in form and punctuation. No major errors were found, only some typos need to be corrected
References:
- Behrens, J.T. Principles and procedures of exploratory data analysis. Methods 1997, 2, 131-160.
- Lan, L.; Yeo, J.H.W. Comparison of computer-key-hold-time and alternating- finger-tapping tests for early-stage Parkinson's disease. PLoS One 2019, 14, e0219114.
- Giancardo, ; Sánchez-Ferro, A.; Arroyo-Gallego, T.; Butterworth, I.; Mendoza, C.S.; Montero, P.; Matarazzo, M.; Obeso, J.A.; Gray, M.L.; Estépar, R.S. Computer keyboard interaction as an indicator of early Parkinson's disease. Sci. Rep. 2016, 6, 34468.
Reviewer 2 Report
1) The article strikes me as novel with potential.
2) “It” statements (eg, “it is important”) should be avoided. So should “there” statements.
3) The first 2 lines of page 3 do not make sense.
4) In line 154 I think the authors mean “shown” rather than “shonwn.”
5) Was a power/sample size determination power?
6) I prefer the first column of tables to be labeled.
Mostly acceptable
Author Response
Reviewer 2:
Comments and Suggestions for Authors
1. The article strikes me as novel with potential.
2. “It” statements (eg, “it is important”) should be So should “there” statements.
Reply: We have rewritten the sentence.
3. The first 2 lines of page 3 do not make sense.
Reply: We have checked the manuscript and these two lines introduced the demographic data.
4. In line 154 I think the authors mean “shown” rather than “shonwn.”
Reply: The typo has been corrected.
5. Was a power/sample size determination power?
Reply: The required sample size was estimated by the following equation:
![]()
Where
σ2 is the estimated population variance
Zα/2 is the critical value of the standard normal distribution at the significance level, as for a confidence level at 95%, αis 0.05, Zα/2 is the critical value for a/2=0.025, which is approximately 1.96.
Zb is the critical value of the standard normal distribution at the desired power level (1-b). Herein we set Zb for b=0.20
δ is the effect size you want to wanted to detect
We found the numbers of the subjects in these three groups fit the minimal requirement to reach a significant power.
To differentiate UPDRS-III, the estimated minimal sample sizes of each group are 9, 5, 4 in the healthy controls, de-novo PD and early PD, respectively.
6. I prefer the first column of tables to be labeled.
Reply: we have labeled the first column of the revised Tables 1-3
Comments on the Quality of English Language Mostly acceptable
Reviewer 3 Report
1. While the dataset was primarily collected from laptop or desktop computers, the conclusion suggests the potential use of smartphones for PD diagnosis without experimental validation. It is crucial to include experimental results to support this claim adequately.
2. The paper lacks a discussion on potential future directions for improvement and further research. Adding a section on future scope could enhance the paper's value and provide insights for researchers.
3. The rationale behind excluding certain available datasets should be clarified. Mentioning the reasons for not considering these datasets would strengthen the methodology's validity.
4. The abstract lacks information about the algorithm used, although it asserts the efficiency of the proposed algorithm in PD patient determination. This information is crucial for readers to understand the approach.
5. The introduction provides the paper's aim, but it lacks details about its contribution and novelty. Describing the unique aspects of the proposed method would better engage readers.
6. The paper mentions a gap in the literature regarding fer, but further literature survey and references are needed to support this claim and provide context.
7. In Figure 1, the symbol 'n' representing the length of the input is undefined, and the timing features are not adequately explained. Considering additional timing features such as di-graph time and statistical features may enhance the feature set and prevent overfitting.
8. The use of 100 seconds for outlier detection lacks justification. Providing reasoning for this specific threshold would enhance the robustness of the analysis.
9. Section 3 should offer a clearer and more detailed identification of the proposed model. Providing a step-by-step description or a flowchart would improve the model's clarity.
10. Demographic data is a crucial aspect of the study and should be included in the Material section for a comprehensive understanding of the sample population.
11. The notable difference between sensitivity and specificity raises concerns about the model's training. Addressing the class imbalance and balancing the datasets would improve the model's suitability for screening purposes.
12. To provide a complete assessment of the proposed model's performance, results for De-novo, early-stage, and combined PD cases should be presented.
13. It is advisable to discuss potential risks associated with the results and the implications of the findings. Addressing limitations and potential biases enhances the paper's credibility.
14. The authors are encouraged to review the recent paper at https://doi.org/10.1016/j.eswa.2023.119522 to gain insights and potentially strengthen their study by incorporating relevant findings and discussions from the referenced work.
15. The following article should be referred with discussion: Imbalanced Ensemble Learning in Determining Parkinson's Disease Using Keystroke Dynamics. Expert Systems with Applications, vol. 217, issue 119522. https://doi.org/10.1016/j.eswa.2023.119522.
16. The same group has many works relevant to this article should be referred.
17. The article needs some obvious better english composition and corrections.
18. More comparative analysis will may make the article more perfect.
19. Resubmit the article after incorporating the corrections
-
Author Response
Reviewer 3:
Comments and Suggestions for Authors
1. While the dataset was primarily collected from laptop or desktop computers, the conclusion suggests the potential use of smartphones for PD diagnosis without experimental It is crucial to include experimental results to support this claim adequately.
Reply: We appreciate the reviewer's feedback. The suggestion regarding the potential use of smartphones for early diagnosis and assessment of disease severity of PD without experimental validation in our conclusion was intended to highlight a potential future direction for research based on the evolving landscape of mobile technology and its applications in healthcare. However, we acknowledge the importance of experimental validation for any new diagnostic approach. In our current study, we primarily utilized data collected from laptop or desktop computers due to the availability of this dataset. However, keystroke dynamics of smartphone has been applied in authentication [1] and assessments for multiple sclerosis [2].
2. The paper lacks a discussion on potential future directions for improvement and further research. Adding a section on future scope could enhance the paper's value and provide insights for researchers. Lines: 348-358
Reply: We appreciate the reviewer's suggestion to include a section on “Future Scope” about our study in the section of Discussion. In our revised manuscript, we included a dedicated paragraph to address this aspect. Lines: 348-358
3. The rationale behind excluding certain available datasets should be clarified. Mentioning the reasons for not considering these datasets would strengthen the methodology's validity.
Reply: Thank you for your comment. We would like to clarify the rationale behind excluding certain available datasets. The primary reason for this exclusion was to maintain data consistency and quality throughout our study. While there were some available datasets, we carefully reviewed each one and excluded those that did not meet our specific criteria for data quality and completeness. These criteria included the availability of essential keystroke parameters, such as typing speed and the number of typed words, which are crucial for our analysis. Incomplete datasets, those with missing or insufficient keystroke data, were excluded as they might not have provided a comprehensive basis for our analysis. we would like to emphasize that our study was designed to provide a comparative analysis using a straightforward approach to typing intervals [3,4].
4. The abstract lacks information about the algorithm used, although it asserts the efficiency of the proposed algorithm in PD patient This information is crucial for readers to understand the approach.
Reply: We appreciate your suggestion regarding including information about the algorithm used in our abstract. However, there is no complicate algorithm in this study, except definition of standard deviation (SD) of natural log of the quotient of two consecutive time intervals, which is declared in the section of “Materials and Methods” Lines 158-159
5. The introduction provides the paper's aim, but it lacks details about its contribution and novelty. Describing the unique aspects of the proposed method would better engage readers.
Reply: Thank you for you for the valuable suggestion. We have provided more details about distinctive aspects of our methodology and study design. Lines 88- 94
6. The paper mentions a gap in the literature regarding fer, but further literature survey and references are needed to support this claim and provide context.
Reply: Thank you for the suggestion. In our revised manuscript, we conducted a more thorough literature review to substantiate the existing gap.
7. In Figure 1, the symbol 'n' representing the length of the input is undefined, and the timing features are not adequately Considering additional timing features such as di-graph time and statistical features may enhance the feature set and prevent overfitting.
Reply: Thank you for the careful checkup. We have correct the typos. Line 143
8. The use of 100 seconds for outlier detection lacks Providing reasoning for this specific threshold would enhance the robustness of the analysis.
Reply: We appreciate the valuable comment. In response, we've made specific improvements to address outlier management in our manuscript. Initially, we set a threshold of 100 seconds to identify and remove extreme outliers based on empirical and visual inspection. Additionally, we utilized the Box plots in the following figure to identify outliers. In the original literature the extreme outliers are defined as those falling outside the range between Q1 - 3xIQR and Q3 + 3xIQR, where IQR = Q3 - Q1 [5]. From the figure we can see that most latency values falling between the lower quartile Q1 and the upper quartile Q3 are in the interval [0,1] second. However, due to the unusual distribution of outliers in the datasets (extreme values more than ±1000), following this outlier definition would have led to the exclusion of too many potentially meaningful samples. To strike a balance, we conducted a thorough visual examination of the raw data, subject-by- subject, and optimized the 12xIQR threshold to maximize the retention of meaningful samples. Consequently, the final data used for analysis are similar, and the results and statistical significance are identical to those obtained using the previous method. We have rewritten the method in the revised manuscript. Lines 149-155.

For example, here are the holding latencies for ID 60 (healthy control) with extreme outliers (A) and the refined data used for analysis after outlier removal (B).

9. Section 3 should offer a clearer and more detailed identification of the proposed Providing a step-by-step description or a flowchart would improve the model's clarity.
Reply: Thank you for the valuable suggestion.
10. Demographic data is a crucial aspect of the study and should be included in the Material section for a comprehensive understanding of the sample population.
Reply: Thank you for the We have put details in the revised manuscript.
11. The notable difference between sensitivity and specificity raises concerns about the model's Addressing the he datasets would improve the model's suitability for screening purposes.
Reply: Thank you for the suggestion. We have made some modifications in the revised manuscript.
12. To provide a complete assessment of the proposed model's performance, results for De-novo, early-stage, and combined PD cases should be presented.
Reply: Thank you for the suggestion. The analyses of combined PD cases, de-novo PD and early PD had been done in previous publications. Novelty of our study is to treated de-novo PD and early PD as two different two groups. The de-novo PD used for assess the efficacy of screening test and the second for assessing the disease severity.
13. It is advisable to discuss potential risks associated with the results and the implications of the findings. Addressing limitations and potential biases enhances the paper's credibility.
Reply: Thank you for the comment. We have rewritten the limitations. Lines 360- 368
14. The authors are encouraged to review the recent paper at https://doi.org/10.1016/j.eswa.2023.119522 to gain insights and potentially strengthen their study by incorporating relevant findings and discussions from the referenced work.
Reply: Thank you for the suggestion.
15. The following article should be referred with discussion: Imbalanced Ensemble Learning in Determining Parkinson's Disease Using Keystroke Dynamics. Expert Systems with Applications, 217, issue 119522.
https://doi.org/10.1016/j.eswa.2023.119522.
Reply: Thank you for the suggestion. We have cited this valuable reference in the limitations of our study.
16. The same group has many works relevant to this article should be referred.
Reply: Thank you for the suggestion.
17. The article needs some obvious better English composition and corrections.
Reply: Thank you for the comment. We have improved it in the revised manuscript.
18. More comparative analysis will may make the article more perfect.
Reply: Thank you for the suggestion. We have added more analyses in the manuscript.
19. Resubmit the article after incorporating the corrections
References:
- El-Kenawy, E.-S.M.; Mirjalili, S.; Abdelhamid, A.A.; Ibrahim, A.; Khodadadi, ; Eid, M.M. Meta-Heuristic Optimization and keystroke dynamics for authentication of smartphone Users. Mathematics 2022, 10, 1-252022, 10, 1-25.
- Lam, H.; Twose, J.; Lissenberg-Witte, B.; Licitra, G.; Meijer, K.; Uitdehaag, B.; De Groot, V.; Killestein, J. The Use of Smartphone Keystroke Dynamics to Passively Monitor Upper Limb and Cognitive Function in Multiple Sclerosis: Longitudinal Analysis. J. Med. Internet Res. 2022, 24, e37614.
- Lan, L.; Yeo, J.H.W. Comparison of computer-key-hold-time and alternating- finger-tapping tests for early-stage Parkinson's disease. PLoS One 2019, 14, e0219114.
- Giancardo, ; Sánchez-Ferro, A.; Arroyo-Gallego, T.; Butterworth, I.; Mendoza, C.S.; Montero, P.; Matarazzo, M.; Obeso, J.A.; Gray, M.L.; Estépar, R.S. Computer keyboard interaction as an indicator of early Parkinson's disease. Sci. Rep. 2016, 6, 34468.
- Behrens, J.T. Principles and procedures of exploratory data analysis. Methods 1997, 2, 131-160.
Round 2
Reviewer 1 Report
I thank the authors for responding, explaining and arguing point by point, appropriately revising the manuscript.
There would still be the following minor fixed, which can be deferred to the final proofreading (but I leave the decision to the editor):
1) Page 6 Line 198: "test between the group between." There is a double "between".
2) Table 2: missing units for the last "SD" parameters (msec?).
3) Table 2 (caption): "SD" should be subscript for HL and IL.
4) line 247: the "#" symbol does not appear in Table 3, so the line can be removed.
5) line 267: the "#" symbol for afTap is not necessary since Table 4 (correlation with UPDRS III) does not cover healthy controls. The symbol and line 267 can be removed.
Author Response
Reviewer 1:
Comments and Suggestions for Authors
I thank the authors for responding, explaining and arguing point by point, appropriately revising the manuscript.
There would still be the following minor fixed, which can be deferred to the final proofreading (but I leave the decision to the editor):
1) Page 6 Line 198: "test between the group between." There is a double "between".
Reply: We thank the reviewer’s careful checkup. The last between has been omitted.
2) Table 2: missing units for the last "SD" parameters (msec?).
Reply: By the definition of SD in Line 158-160, SD is the standard deviations of the natural log of the quotient of two consecutive time intervals. Therefore, the SD parameters do not have units.
3) Table 2 (caption): "SD" should be subscript for HL and IL.
Reply: We thank the reviewer’s careful checkup. The SDs have been corrected.
4) line 247: the "#" symbol does not appear in Table 3, so the line can be removed.
Reply: We thank the reviewer’s careful checkup. The missed # has been added in Table 3 as afTap#.
5) line 267: the "#" symbol for afTap is not necessary since Table 4 (correlation with UPDRS III) does not cover healthy controls. The symbol and line 267 can be removed.
Reply: We appreciate the reviewer’s careful checkup. Table 4 has been corrected as the comment.